# ZIKV Strains Elicit Different Inflammatory and Anti-Viral Responses in Microglia Cells

**DOI:** 10.3390/v15061250

**Published:** 2023-05-26

**Authors:** Fernanda Bellaniza Caminha de Oliveira, Vanessa Paola Alves Sampaio de Sá Freire, Sharton Vinicius Antunes Coelho, Lana Monteiro Meuren, Julys da Fonseca Palmeira, Ana Luísa Cardoso, Francisco de Assis Rocha Neves, Bergmann Morais Ribeiro, Gustavo Adolfo Argañaraz, Luciana Barros de Arruda, Enrique Roberto Argañaraz

**Affiliations:** 1Laboratory of Molecular Neurovirology, Department of Pharmacy, Faculty of Health Science, University of Brasília, Brasília 70910-900, DF, Brazil; febellaniza@hotmail.com (F.B.C.d.O.); vanessa.safreire3@gmail.com (V.P.A.S.d.S.F.); julys.palmeira@hotmail.com (J.d.F.P.); gustaad2003@yahoo.com.br (G.A.A.); 2Laboratório de Genética e Imunologia das Infecções Virais, Departamento de Virologia, Instituto de Microbiologia Paulo de Goes, Universidade Federal do Rio de Janeiro, Rio de Janeiro 21941-902, RJ, Brazil; shartonvinicius@micro.ufrj.br (S.V.A.C.); meuren.lana@gmail.com (L.M.M.); arruda@micro.ufrj.br (L.B.d.A.); 3Centre for Neuroscience and Cell Biology, University of Coimbra, 3004-517 Coimbra, Portugal; cardoso.alc@gmail.com; 4Laboratory of Molecular Pharmacology, Faculty of Health Science, University of Brasília, Brasilia 70910-900, DF, Brazil; nevesfar@gmail.com; 5Laboratory of Bacuolovirus, Cell Biology Department, University of Brasilia, Brasilia 70910-900, DF, Brazil; bergmann@unb.br

**Keywords:** Zika virus, miRNA, PPAR-γ, microglia, inflammatory response

## Abstract

In recent years, the Zika Virus (ZIKV) has caused pandemic outbreaks associated with a high rate of congenital ZIKV syndrome (CZS). Although all strains associated with worldwide outbreaks derive from the Asian lineage, the reasons for their enhanced spread and severity are not fully understood. In this study, we conducted a comparative analysis of miRNAs (miRNA-155/146a/124) and their cellular targets (SOCS1/3, SHP1, TRAF6, IRAK1), as well as pro- and anti-inflammatory and anti-viral cytokines (IL-6, TNF-α, IFN-γ, IL-10, and IFN-β) and peroxisome proliferator-activated receptor γ (PPAR-γ) expression in BV2 microglia cells infected with ZIKV strains derived from African and Asian lineages (ZIKV_MR766_ and ZIKV_PE243_). BV2 cells were susceptible to both ZIKV strains, and showed discrete levels of viral replication, with delayed release of viral particles without inducing significant cytopathogenic effects. However, the ZIKV_MR766_ strain showed higher infectivity and replicative capacity, inducing a higher expression of microglial activation markers than the ZIKV_PE243_ strain. Moreover, infection with the ZIKV_MR766_ strain promoted both a higher inflammatory response and a lower expression of anti-viral factors compared to the ZIKV_PE243_ strain. Remarkably, the ZIKK_PE243_ strain induced significantly higher levels of the anti-inflammatory nuclear receptor—PPAR-γ. These findings improve our understanding of ZIKV-mediated modulation of inflammatory and anti-viral innate immune responses and open a new avenue to explore underlining mechanisms involved in the pathogenesis of ZIKV-associated diseases.

## 1. Introduction

Zika virus (ZIKV) is an arbovirus (arthropod-borne virus) from the *Flaviviridae* family [1]. It was first identified in Africa in 1947, and two different lineages have been classified: the African lineage and the Asian lineage [2]. While the African lineage has not caused widespread concern, the Asian lineage has led to increasing public health worries due to its epidemic outbreaks in Micronesia (2007) [3], French Polynesia (2013) [4], and Brazil (2015) [5] as well as other outbreaks reported in 2019 [6].

Studies conducted on the brains of fetuses and newborns infected by Asian lineage-derived strains in Brazil have shown devastating injuries to the central nervous system (CNS), including microcephaly, hydrocephalus, necrosis, periventricular and cortical calcification, diffuse astrogliosis and microglia activation. These symptoms collectively constitute what is known as congenital Zika syndrome (CZS) [7,8,9]. In adult patients, the disease caused by the Zika virus typically manifests as asymptomatic or mild symptoms, including fever, rash, joint pain, myalgia, and conjunctivitis. However, approximately 10% of patients may experience varying degrees of neurological complications, such as headache, Guillain–Barre syndrome (GBS), meningoencephalitis, or acute myelitis [10,11]. Recent studies have indicated that macrophage infection in lymph nodes is a critical step in the early stages of ZIKV dissemination from the skin to the blood stream [12]. Moreover, virus particles or RNA and anti-ZIKV antibodies have been detected in the brain and cerebrospinal fluid of fetuses, stillborns, and adult patients, indicating the virus’s capacity to invade and replicate within the central nervous systems [13,14].

Different strains of ZIKV derived from the African and Asian lineages (MR766 and PE243 strains, respectively) have been found to exhibit distinct differences in their sequences and cytopathic effects in various experimental models involving cell culture, mice and non-human primates [15,16,17,18,19]. Notably, infection with ZIKV_MR766_ has been associated with higher rates of animal death, increased virus replication, and elevated levels of TNF-α, IL-6, and IL-1β in the brain [20]. Although African strains, such as ZIKV_MR766_, exhibit higher infectivity and replication capacities in human neural progenitor cells (hNPCs) and cause significant defects in cell proliferation and apoptosis, no reports of CZS have been associated with this lineage [21].

The host’s innate immune response against ZIKV infection is initiated by the recognition of viral RNAs by pattern recognition receptors (PRRs) in host cells. PRRs relevant to flavivirus infection include Toll-like receptor (TLR), TLR3, and TLR7/8 [22], as well as RIG-I-like receptors (RLRs), including Retinoic acid-Inducible Gene I (RIG-I) and Melanoma Differentiation-Associated gene 5 (MDA5), which are involved in early and late innate immune responses, respectively [19,23,24]. Upon activation by single or double-stranded RNA, TLR3 and TLR7/8 interact with their adapter proteins, TIR-domain-containing adaptor-inducing interferon-β (TRIF) and myeloid differentiation primary response 88 (MyD88), respectively [25]. Activation of TRIF in the downstream signaling pathway TRIF/RIP1/TRAF3/TBK-1/IKKe and IRF-3/7 induces type I interferon transcription. On the other hand, activation of TLR7-8/IL-1R-associated kinase1(IRAKs)/TNFR-associated factor 6 (TRAF6) and IKKα/β leads to NF-κB activation and expression of pro-inflammatory cytokines [26].

The importance of the host’s IFN innate immune response against ZIKV infection is underscored by the increased susceptibility of IFNAR-deficient mice [18] and the protective effect of type III IFN against ZIKV infection in human placental cells [27]. However, ZIKV has developed various mechanisms to evade the host’s IFN responses in order to replicate efficiently [28,29,30].

Virus sensing also triggers the expression of microRNAs (miRNAs), which can target transcripts involved in different TLR signaling pathways. Modulation of miRNA expression is an important regulatory mechanism for the timely and appropriate control of the pro-inflammatory response [31,32]. Although significant progress has been made in understanding the molecular interaction between flaviviruses and the host cell, the molecular role of the host immune cell’s miRNAs in the innate immune response and viral immune evasion during ZIKV infection remains incompletely understood [33,34].

Various pathological findings in ZIKV-infected fetuses have raised questions about potential glial disturbances [7,34]. Astrocytes are susceptible to ZIKV infection, with high viral production suggesting they may work as viral reservoirs [35]. Microglia cells are important sources of inflammatory factors associated with neuronal pathologies, leading to pro-inflammatory cytokines production, such as TNF-α, IL-6, IL-1β, IL-12, and nitric oxide (NO) [36,37]. ZIKV infection of microglia has been associated with altered expression of metabolites involved in various functions, including neuronal differentiation, apoptosis, virion structure, viral replication, and secretion of inflammatory mediators, which may contribute to SNC inflammation [38]. Peroxisomal Proliferator-Activated Receptors (PPARs) play a crucial role in reducing inflammatory responses in microglia and astrocytes, blocking amyloidogenic pathways, inducing neuronal differentiation and neurite growth, inhibiting apoptosis, oxidative damage, inflammation, and mitochondrial dysfunction, thus exerting neuroprotective effects [39,40,41]. PPAR-γ has been shown to be expressed in microglia, including murine cell line BV2, mediating anti-inflammatory activity and inducing microglial polarization of the M2-immunosuppression phenotype [42,43]. PPARs also participate in lipid metabolism and peroxisome biogenesis, which are key factors in anti-viral signaling [44,45]. Despite these findings, the molecular role of PPARs in the immune response against ZIKV and their ability to counteract viral immune evasion mechanisms remains largely unexplored.

Here, we describe the remarkable differences in microglial cell activation, viral replication, and inflammatory and anti-viral responses against ZIKV strains derived from African and Asian lineages. The ZIKV_MR766_ strain promoted higher cell activation, higher inflammatory response, and lower expression of some anti-viral factors in an early stage of infection. In contrast, the Asian lineage-derived ZIKV_PE243_ strain induced significantly higher levels of PPAR-γ and lower levels of pro-inflammatory cytokines IL-6 and TNF-α.

## 2. Materials and Methods

### 2.1. Virus and Cells

The Brazilian isolate of ZIKV, strain PE243 (ZIKV_PE423_, Genbank accession number KX197192) was kindly provided by Dr. Ernesto T.A. Marques Jr (Centro de Pesquisas Aggeu Magalhães, FIOCRUZ, PE, Brazil; and Center for Vaccine Research, University of Pittsburgh, Pittsburgh, PA, USA). The ancient ZIKV strain MR766 (ZIKV_MR766_; ATCC VR1838) was donated by Dr. Amilcar Tanuri (Instituto de Biologia, Federal University of Rio de Janeiro, UFRJ, RJ, Brazil). Both ZIKV isolates were propagated in C6/36 cells, and virus titers were determined by plaque assay using Vero cells, as previously described [46]. The supernatants obtained from non-infected C6/36 cells cultured in the same conditions were used as mock controls.

The C6/36 cell line (ATCC^®^ CRL-1660^™^) was cultured in Leibovitz (L-15) medium (Life Technologies, Carlsbad, CA, USA), supplemented with 10% of tryptose phosphate broth, 0.75% sodium bicarbonate, 0.2% of l-glutamine (Sigma-Aldrich, St. Louis, MO, USA), and 10% FBS (Life Technologies), and maintained at 28 °C. Vero cells (ATCC^®^ CCL81) were cultured in DMEM supplemented with L-glutamine and 5% fetal bovine serum (Life Technologies, Grand Island, NY, USA) and maintained at 37 °C with 5% CO_2_. The C57BL/6 mouse brain microglia cell line-BV2 (BCRJ ref. number 0356) was cultured in RPMI-1640 medium (LGC), supplemented with 2.05 mM L-Glutamine, 25 mM HEPES buffer, 2 g/L sodium bicarbonate, 10% FBS, and maintained at 37 °C with 5% CO_2_.

### 2.2. BV2 Cells Infection and Treatment

BV2 cells were cultured with ZIKV_PE243_ or ZIKV_MR766_, with the indicated MOIs (from 0.1 to 10) for 2 h at 37 °C in a 5% CO_2_ atmosphere under gentle shaking. As a control, the cells were incubated with a supernatant of non-infected C6/36 cells (mock-treated). The cells were washed with PBS and then with medium, and the medium was collected as the zero-infection time (input) before being cultured with the complete medium. After incubation at different time points, cells and supernatants were harvested, and virus infection, cell survival, and cytokine secretion were evaluated as described below. As a positive control for inflammation, cells were treated with LPS [0.1 mg/mL] for 18 hpi and 1 mM ATP for the last 30 min.

### 2.3. Cell Viability Assays

BV2 cells were infected with ZIK_PE243_ or ZIKV_MR766,_ and determination of cell viability was carried out using XTT 2,3-Bis-(2-Methoxy-4Nitro-5-Sulfophenyl)-2H-Tetrazolium-5-Carboxanilide (XTT) (Sigma-Aldrich) with MOIs of 0.1, 1, and 5, at different time points. Cells were incubated with XTT solution for 2–4 hpi, and metabolization was evaluated by spectrophotometry at 450 nm OD. A positive control using 1% Triton X100 was used included.

### 2.4. Analysis of BV2 Infection by Immunofluorescence, Quantitative Reverse Transcription PCR (qRT-PCR), and Plaque Assay

BV2 cells were either mock-treated or infected with ZIKV_PE243_ or ZIKV_MR766_ with different MOIs, as mentioned before. After 24, 48, or 72 hpi, cells were fixed with 4% paraformaldehyde in PBS for 20 min, washed in PBS twice, and permeabilized with 0.1% Triton X-100 (Sigma Aldrich) plus 3% bovine serum albumin (BSA—Sigma Aldrich) for 25 min. Then, cells were pre-stained with anti-dsRNA J2 antibody (0.1 mg/mL, Ref: RNT-SCI-10010200, Jena Bioscience GmbH, Jena, Germany), followed by anti-mouse IgG conjugated to PE (1 µg/mL, Ref: A32744, Invitrogen, Waltham, MA, USA), and DAPI. Cells were analyzed by immunofluorescence, using a Zeiss Axioimager d2 microscope and 63× magnitude.

The kinetics of viral replication was evaluated by Real-Time Quantitative PCR (RT-qPCR) and plaque assay at 0, 8, 16, 24, 48, 72, and 96 hpi with an MOI of 1. Cells and supernatants were harvested, and RNA was isolated using TRIZOL reagent (Life Technologies), according to the manufacturer’s instructions. Treatment with DNAse I (Ambion, Thermo Fischer, Waltham, MA, USA) was performed to prevent genomic DNA contamination, and cDNA strands were synthesized using a High-Capacity cDNA Archive Kit (Applied Biosystems, Waltham, MA, USA), according to the manufacturer’s instructions. The cDNAs were subjected to quantitative RT-qPCR for detection of viral RNA, carried out in a StepOne Plus Real-time PCR system and with Taqman Master Mix Reagents (Applied Biosystems), using primers and probe specific for the protein E sequence, as previously described [47]. cDNA obtained from virus samples ranging from 75,000 to 0.75 PFU/mL were used to construct a standard curve for estimating the genome copy number of ZIKV (RNA equivalent). The values obtained were normalized from the subtraction by the input values. The concentration of virus infectious particles (PFU/mL) in the supernatants obtained from BV2-infected cells was determined by plaque assay using Vero cells, as previously described [46].

### 2.5. Evaluation of BV2 Cells Activation

BV2 cells were mock-treated, infected with ZIKV_PE243_ or ZIKV_MR766_, with an MOI of 1, or stimulated with LPS + ATP as a positive control. After 24, 48, and 72 hpi, cells were fixed and permeabilized as described previously. Then, the cells were incubated overnight with the primary anti-Iba1 antibody at 1:500 (Wako, code 019-19741), anti-CD68 antibody at 1:500 (Bio-Rad, MC1957, Hercules, CA, USA), or with anti-MHC II conjugated with phycoerythrin—PE (eBioscience, San Diego, CA, USA, ref. 12-5322-81). The cells were then stained with the respective anti-IgG secondary antibodies conjugated to AlexaFluor488 or AlexaFluor647. To stain cell nuclei, a mounting medium containing 4,6-diamidino-2-phenylindole (DAPI) (blue) (Vectashield—Vector Laboratories, Mowry Ave Newark, CA, USA) was used. Fluorescence microscopy was performed simultaneously with the same antibody solutions, and all micrographs were taken with the same settings using a Zeiss LSM 710 confocal microscope equipped with a 63× objective and digital imaging system (Roper Scientific camera and Zen (black edition) software).

### 2.6. Quantification of microRNAs and Inflammatory Mediator’s Expression in ZIKV-Infected BV2 Cell Line

BV2 cells were infected with ZIKV_PE243_ or ZIKV_MR766_, as previously described. After 12, 24, 48, and 72 hpi, the mRNAs and miRNAs from infected cells were purified with TRIZOL reagent (Life Technologies), as described previously, and the mirVana PARIS kit (Protein and RNA Isolation System—Invitrogen), respectively, according to the manufacturer’s instructions. To ensure optimum performance of extraction assays, the extracted miRNA samples were run on RNAse-free agarose gel to ensure they were intact and not contaminated with DNA. For the infection experiment with Rosiglitazone treatment, BV2 cells were plated and infected by both isolates, but immediately after infection, the cells were treated with Rosiglitazone at three different concentrations: 10^−5^, 10^−6^, and 10^−7^ M. After 48 h of treatment, mRNA extraction was performed with the GenElute Mammalian Total RNA Miniprep kit (Sigma-Aldrich). mRNA and miRNA quantification was carried out in a Quantus fluorimeter (Promega, Madison, WI, USA) and normalized to an amount of 2 μg for mRNA and 10 ng for miRNA, according to the manufacturer’s recommendations.

The cDNAs of miR-155, miR-146a, miR-124, and miR-132 were synthesized using TaqMan^®^ MicroRNA Reverse Transcription Kit mix (Applied Biosystems), and microRNA-specific primers, following the manufacturer’s instructions, with the following parameters: 16 °C for 30 min, 42 °C for 30 min, 85 °C for 5 min and 4 °C. Subsequently, quantification of each miRNA expression was performed with specific microRNA probes for each target, in a StepOne Plus Real-Time PCR System (Applied Biosystems). The values obtained from gene expression were normalized from the values acquired from the small endogenous nuclear RNA U6 and negative infection control (mock) by ∆∆CT calculation [48].

The cDNAs for the target genes were synthesized using a High-Capacity cDNA kit (Applied Biosystems), following the manufacturer’s instructions, as previously mentioned. The expression of SOCS1/3, TRAF6, IRAK1, TNF-α, IL-6, IFN-γ, IL-10, and IFN-β genes was measured by SYBR Green Master Mix (Applied Biosystems) in a StepOne Plus Real-Time PCR System (Applied Biosystems). The following primers were used: SOCS1 sense: 5′-CTGCGGCTTCTATTGGGGAC-3′, SOCS1 antisense: 5′-AAAAGGCAGTCGAAGGTCTCG-3′; SOCS3 sense: 5′-GCTCCAAAAGCGAGTACCAGC-3′; SOCS3 antisense: 5′-AGTAGAATCCGCTCTCTTGCAG-3′; TRAF6 sense: 5′-AAGATTGGCAACTTTGGATG-3′, TRAF6 antisense: 5′-GTGGGATTGTGGGTCGCTG-3′; IRAK1 sense: 5′-ATCAGGCTTTTTCCCAGGCTT-3′ IRAK1 antisense: 5′-CACCTGATGCCTTTGGGCTA-3′ TNF-α sense: 5′-CCCTCACACTCAGATCATCTTCT-3′, TNF-α antisense: 5′-GCTACGACGTGGGCTACAG-3′; IL-6 sense: 5′-TAGTCCTTCCTACCCCAATTTCC-3′, IL-6 antisense: 5′-TTGGTCCTTAGCCACTCCTTC-3′; IFN-γ sense: 5′-CGTCATTGAATCACACCTG-3′, IFN-γ antisense: 5′-GGTTGTTGACCTCAAACTTG-3′; IL-10 sense: 5′-GCGCTGTCATCGATTTC-3′, IL-10 antisense: 5′-GTCAAATTCATTCATGGCC-3′; IFN-β sense: 5′-TAGCACTGGCTGGAATGAGA-3′, IFN-β antisense: 5′-TCCTTGGCCTTCAGGTAATG-3′; HPRT1 sense: 5′-CCCTGGTTAAGCAGTACAGC-3′, HPRT1 antisense: 5′-ATCCAACAAAGTCTGGCCTG-3′; GAPDH sense: 5′-AAGGGCTCATGACCACAGTC-3′; GAPDH antisense: 5′-CAGGGATGATGTTCTGGGCA-3′. The gene expression of CD68, SHIP1, PPAR-γ, was measured by TaqMan^®^ commercial probes from Thermo Fisher: CD68 (Mm04411920_m1), SHIP1(Mm01290317_m1), and PPAR-γ (Peroxisome Proliferator Activated Receptor Gama; ID: Mm01301752_m1). The relative expression values of the targets in infected cells were normalized against the values obtained from endogenous cell control (HPRT1) and negative infection control (mock) using the ∆∆CT calculation and corrected with the efficiency value [49].

### 2.7. Western Blotting

The total protein of BV2 cells infected with either ZIKV_PE243_ or ZIKV_MR766_ was extracted using RIPA buffer (Sigma-Aldrich) and quantified using the Dye Reagent Concentrate (Bio-Rad Protein Assay), following the manufacturer’s instructions. Protein extracts were mixed with 2× Laemmli sample buffer (Bio-Rad, Hercules, CA, USA) and subjected to polyacrylamide gel electrophoresis (SDS-PAGE), followed by transfer to a nitrocellulose membrane (Thermo Fisher Scientific Inc.). The membranes were blocked using 5% nonfat dried milk diluted in 1× Tris Buffered Saline (TBS) with 1% Tween-20. Primary antibodies against PPAR-γ (Thermo Fisher Scientific Inc., PA3-821A) and SOCS1 (Boster Biological Technology Co. Ltd, Pleasanton, CA, USA, PA1074) were added to the membranes, and then secondary antibodies against mouse (Cell Signaling Technology, 7076S, Danvers, MA, USA) and rabbit (KPL, 04-15-06) were used. The membranes were further incubated with HRP-conjugated secondary antibodies (Thermo Fisher Scientific Inc.).

### 2.8. Statistical Analysis

Data were analyzed using the GraphPad Prism 8.0.0 software (GraphPad Software, San Diego, CA, USA). Comparisons among groups were performed by one-way ANOVA, followed by Tukey’s test; *p* < 0.05 were considered statistically significant.

## 3. Results

### 3.1. African Lineage-Derived ZIKV_MR766_ Strain Infects and Showed Higher Replication Levels in BV2 Microglial Cells Than Asian Lineage-Derived ZIKV_PE243_ Strain

Firstly, we investigated the susceptibility and permissiveness of murine BV2 microglial cells to ZIKV. To assess this, BV2 cells were infected with ZIKV_PE243_ or ZIKV_MR766_ at MOIs of 1 or 10, and the expression of ZIKV dsRNA was analyzed by immunofluorescence. ZIKV_PE243_-infected BV2 cells exhibited a lower infection level compared to ZIKVMR766-infected cells when using an MOI of 10 (Figure 1a), whereas, with an MOI of 1, the signal infection was not clearly detected.

Additionally, we evaluated the viral replication kinetics by measuring the intracellular viral RNA levels by RT-qPCR and the release of infectious particles by PFU assay, with different MOIs, and at different time points post-infection. We observed that virus RNA levels were slightly higher in the cell lysates obtained from ZIKVMR766-infected cells compared to ZIKV_PE243_-infected cells (Figure 1b,c). Furthermore, infection with both strains was associated with a delayed virus release, but ZIKV_MR766_ replication resulted in the release of a higher infectious virus particles (PFU), in comparison to ZIKVPE243 (Figure 1d). These results support previous findings obtained by our group in human brain microvascular endothelial cells (HBMEC), which also showed higher infectivity of the ZIKV_MR766_ strain [14]. Moreover, we investigated whether ZIKV infection affected BV2 microglial cells’ physiology and survival. XTT metabolization assays were conducted in BV2 cells infected with ZIKV_PE243_ and ZIKV_MR766_ at different time points and MOIs until control cell cultures began to deteriorate. We did not observe significant cytopathic effects in the cultures, and no significant changes in XTT metabolization were observed in any condition tested, indicating no cell death (Figure 1e,f). The low cytopathic effect observed may be attributed to the low susceptibility of BV2 microglia cells to ZIKV.

### 3.2. African Lineage-Derived ZIKV_MR766_ Strain Induces Greater BV2 Microglia Cells Activation Than ZIKV_PE243_

Considering that the levels of viral RNA detected after infection with MOIs of 1 and 5 at 24 hpi were similar, which might indicate a replication threshold in this system, we chose an MOI of 1 to proceed with the analysis of the effect of ZIKV in BV2 microglial cells activation. Thus, we examined the expression of activation markers: namely Iba1, MHC-II, and CD68, using immunofluorescence at various time points after infection. We used LPS + ATP stimulation as a positive control (Figure 2a). The expression of the Iba1 protein showed a similar pattern upon infection with both strains, with a slight increase in a time-dependent manner. However, ZIKV_MR766_ infection induced higher MHC-II and CD68 expressions when compared to ZIKV_PE243_ infection. Furthermore, the analysis of CD68 mRNA expression showed a time-dependent increase, with higher levels observed in ZIKV_MR766_-infected cells compared to mock-infected cells (*p* < 0.001 and *p* < 0.01, 48 and 72 hpi, respectively) (Figure 2b).

### 3.3. ZIKV_PE243_ and ZIKV_MR766_ Differentially Modulate Pro- and Anti-Inflammatory Cytokines and Anti-Viral Factors’ Expression in BV2 Microglial Cell

In a previous study, we demonstrated that both ZIKV_MR766_ and ZIKV_PE243_ could induce HBMEC cells to produce different levels of chemokines and pro-inflammatory cytokines [14]. In this study, we evaluated the pattern of inflammatory cytokines in microglial BV2 cells infected with the same ZIKV strains. Interestingly, ZIKV_MR766_ infection stimulated higher TNF-α and IL-6 levels than mock-infected cells at the initial stage of infection, i.e., 12 hpi (*p* < 0.001 and *p* = 0.4567, respectively), whereas ZIKV_PE243_ failed to induce significant changes (Figure 3a,b). Additionally, we observed a greater increase in pro-inflammatory cytokine IFN-γ expression in cells infected with both strains, at 48 and 72 hpi (*p* < 0.001), with higher levels detected in ZIK _MR766_-infected cells (Figure 3c). Conversely, the expression of the immunomodulatory cytokine IL-10 gene was found to increase in ZIKV-infected cells, starting at 24 hpi, with slightly higher levels in cells infected with ZIKV_PE243_ (Figure 3d).

These results indicate that ZIKV_MR766_ can induce higher levels of inflammation than ZIKV_PE243_ in BV2 microglial cells during the early stages of infection. To further investigate the anti-viral response, we examined the expression of type I interferon (IFN-β) and type III interferon (IFN-λ2/3) by RT-qPCR. Both viral strains induced significant expression of IFN-β, with a time delay starting at 24 hpi (Figure 3e). Interestingly, at 24 hpi, ZIKV_PE243_ induced higher levels of IFN-β expression than the ZIKV_MR766_ strain (*p* < 0.001 compared to mock-infected cells and between both strains). However, we did not detect the expression of the INF-λ2/3 gene in BV2 microglial cells infected with either ZIKV strain.

### 3.4. ZIKV_PE243_ and ZIKV_MR766_ Strains Differentially Modulate the Pro-Inflammatory miRNA-155 and Anti-Inflammatory and Anti-Innate Immune Response Factors in BV2 Microglial Cells

To gain a deeper understanding of the molecular mechanisms underlying the differential inflammatory and anti-viral response to ZIKV, we investigated the expression kinetics of key microglia-associated miRNAs and related pro- and anti-inflammatory factors. We began by assessing miRNA-155, which is known to downregulate anti-innate immune response and anti-inflammatory host transcripts, including suppressor of cytokine signaling 1 and 3 (SOCS1/3) and Src homology 2-containing inositol phosphatase 1 (SHIP1). As expected, LPS + ATP treatment induced a marked increase in miRNA-155 expression, indicative of an inflammatory profile (Figure 4a). ZIKV infection led to a gradual increase in miRNA-155 expression between 12 and 24 hpi, followed by a decline at 48 hpi and a subsequent significant increase at 72 hpi. Notably, ZIKV_MR766_-BV2-infected cells exhibited earlier and higher upregulation of miRNA-155 compared to ZIKV_PE243_-infected cells in relation to mock-infected cells (*p* < 0.05 at 24 hpi) (Figure 4a).

The analysis of *socs1*/*ship1* gene expression, cellular targets of miRNA 155, showed that both ZIKV strains induced a significant early increase at 12 hpi, followed by a gradual decline at later times (Figure 4b,e). ZIKV_PE243_-infected cells showed significantly higher mRNA expressions levels than ZIKV_MR766_-infected cells at 12 hpi (*p* < 0.001 on both genes) and 24 hpi (*p* < 0.01 and *p* < 0.001 for *socs1* and *shpi*, respectively) (Figure 4b,e). Representative Western blot analysis and quantification data confirmed the higher ZIKV_PE243_-mediated SOCS1 protein expression at the early phase of ZIKV infection (12 and 24 hpi) (Figure 4c). Regarding the expression of SOCS3, there was an increase in expression over time, without significant difference in the early stages of infection (12 and 24 hpi), with a greater increase in cells infected with the ZIKV_MR766_ strain at later stages (*p* < 0.05 in both) (Figure 4d).

### 3.5. ZIKV_PE243_ and ZIKV_MR766_ Differentially Modulate the Anti-Inflammatory miRNAs-146a/124 and Inflammatory Mediators’ Expressions

We also investigated the expression of anti-inflammatory miRNAs 146a and 124, known to regulate the expression of TRAF6, IRAK-1, and MAVS essential adaptor proteins required for type I IFN production, as well as the pro-inflammatory proteins TNF-α and IL-6 [50,51,52]. At 24 hpi, BV2 cells infected with both ZIKV strains showed a significant increase in miRNAs 146a and 124, with higher levels observed in cells infected with ZIKV_MR766_ (Figure 5a,b). While the expression levels of miRNA146a decreased at later times, miRNA124 levels remained high until 48 hpi in ZIKV_PE243_-infected cells (*p* < 0.01). The transcript expression level of TRAF6 inflammatory protein increased remarkably at 12 hpi with both strains compared to mock-infected cells (*p* < 0.001 in both strains), followed by a decrease to levels similar to mock at later time points (Figure 5c). IRAK1 expression showed a delayed pattern, starting at 24 hpi and reaching significant differences from mock-infected cells only at 48 hpi (*p* < 0.05 for ZIKV_MR766_) and 72 hpi (*p* < 0.001 for ZIKV_PE243_) (Figure 5d).

### 3.6. BV2 Cells Infected with ZIKV_MR766_ Display Lower PPAR-γ Expression Levels

Since PPAR-γ is known to play a crucial role in the anti-inflammatory process in glial cells by regulating lipid metabolism and producing IL-6 and TNF-α inflammatory cytokines [39,40,53], we examined PPAR-γ expression in BV2 infected cells with both ZIKV strains. Our findings indicate that ZIKV infection led to a time-dependent increase in PPAR-γ expression. Notably, significant PPAR-γ expression was detected in ZIKV_PE243_-infected cells as early as 12 hpi, with statistically significant differences between strains at 24 and 48 hpi (*p* < 0.01 and *p* < 0.001, respectively). In contrast, PPAR-γ induction by ZIKV_MR766_ was lower and occurred later, starting at 48 hpi (Figure 6a). Our negative controls included LPS + ATP and mock-treated cells, while mRNA from murine adipocyte cells served as a positive control. Western blot analysis and quantification data also demonstrated higher PPAR-γ protein expression in microglial BV2-ZIKV_PE243_ infected cells (Figure 6b). We did not observe any detectable PPAR-α gene expression in BV2 microglial cells infected with either ZIKV strain. As we noticed a difference in early TNF-α and IL-6 production after infection with the two strains (Figure 3), we examined whether PPAR-γ activation could play a role in the expression of these cytokines. To explore this possibility, we infected BV2 cells and immediately treated them with various concentrations of Rosiglitazone, a PPAR-γ agonist drug. After 48 hpi, we measured the expressions of IL-6 and TNF-α. We used LPS-stimulated cells as a positive control, while the addition of Rosiglitazone to mock-treated cells served as an internal control for the assay. Treatment of BV2-infected cells with all concentrations of the PPAR-γ agonist resulted in a significant reduction in the expression levels of both IL-6 and TNF-α, compared to infected cells without treatment (Figure 6c,d).

Table 1 summarizes and contrasts the expression levels of inflammatory, anti-inflammatory, and anti-viral factors induced in BV2 cell lines infected by both ZIKV strains, ZIKV_PE243_ and ZIKV_MR766_. These data sets suggest that the Asian and African ZIKV strains induce differential innate immune responses, which may contribute to different clinical outcomes.

## 4. Discussion

The distinct virulence and pathogenesis between African and Asian lineages of ZIKV present a unique opportunity to investigate the molecular mechanisms associated with disease and protection. Researchers are particularly interested in studying how viral sequence differences contribute to virulence, tissue tropism, pathology, and immune evasion [19]. Infection with the Asian lineage of ZIKV has been linked to CNS alterations in newborns exposed to the virus during the prenatal period, as well as meningoencephalitis in adults [13,54]. Since glial cells play a critical role in regulating neuroinflammatory processes during Zika infection of the CNS [36,38], we conducted a comparative analysis of pro- and anti-inflammatory, and anti-viral factors and miRNAs in microglial cells infected with ZIKV strains from African and Asian lineages, at both early and late stages of infection.

ZIKV_MR766_ or ZIKVP_E243_ do not induce cytopathic effects in BV2 microglial cells due to their low levels of infection, replication, and delayed release of viral particles. In contrast, other cell types, such as human brain microvascular endothelial cells HBMECs, neuronal stem cells, brain organoids, and other microglial cell lines, exhibit higher levels of infection and replication [14,15,38,53]. This discrepancy may be attributed to the virus’s tropism for different CNS cell types [38,55] as well as the murine origin of the BV2 microglial cell line, which may have led to the virus counteracting the human, but not the mouse, interferon response [30,56]. Nonetheless, ZIKV_PE243_ demonstrates lower infectivity and replication efficiency compared to the ZIKV_MR766_ strain, indicating that differences in the genomic sequences between the two ZIKV strains may impact the virus’s replication capacity. Previous studies have demonstrated that ZIKV_MR766_ has a higher infectivity and replication rate in hNPCs, despite being anciently replicated in mouse brains, suggesting replicative adaptation in mouse cells and high virulence as an intrinsic characteristic of the African ZIKV strain [57,58]. In contrast, ZIKV strains derived from Asian lineage have a lower virulence, causing mild and persistent infection of neural cells and leading to congenital malformations.

The expression of microglial activation markers, including Iba1, CD68, and MHC-II after infection, along with the low replication levels and absence of CPE, suggests that microglia may serve as a viral reservoir and a source of inflammation in the CNS [18]. The high levels of MHC-II and CD68 expression induced by the ZIKV_MR766_ strain are in line with its higher infectivity. The lack of correlation between CD68 mRNA expression and membrane protein expression by immunofluorescence may be attributed to the fact that the CD68 protein is predominantly expressed in the lysosomal membranes rather than the plasma membrane. CD68 gene expression may be regulated by negative feedback mechanisms triggered upon reaching a certain expression threshold in the lysosomal membrane [59]. The ZIKV_MR766_ strain exhibited greater infectivity and replicative capacity, as reflected in the induction of pro-inflammatory cytokines expression (TNF-α, IL-6, and IFN-γ). Higher levels of IFN-γ expression induced by ZIKV_MR766_ may be due to the NS5 protein of this strain having a greater capacity to induce increased STAT1-SAT1 homodimerization [60]. Additionally, the ZIKV_MR466_ strain’s higher INF-γ levels may contribute to the expression of pro-inflammatory cytokines in the later stages of infection, which could facilitate viral spread. The expression of inflammatory cytokines induced by ZIKV strains was inversely correlated with the expression levels of the anti-inflammatory cytokine IL-10, consistent with a negative feedback loop [61]. Furthermore, the slightly higher production of IL-10 in response to ZIKV_PE243_ in the early stages of infection (12 and 24 hpi) may be more harmful than beneficial to the pregnancy outcomes, as it could promote viral persistence and dampen host defense mechanisms.

The delayed response of IFN-β in BV2 cells to ZIKV infection may be attributed to multiple factors, including inefficient detection of viral RNA and/or suppression of the RNA-mediated anti-viral signaling pathway by viral proteins, such as NS1, NS4, and NS5 [19,29]. Moreover, the lower expression of IFN-β in BV2 cells infected with ZIKV_MR766_, particularly at 24 hpi, may be linked to the higher efficiency of NS5 viral protein in antagonizing the host protein TBK1, resulting in reduced phosphorylation of the TBK1/TRAF6 complex and IRF3 [62]. In contrast, the lack of IFN-λ response by BV2 microglial cells is consistent with the role of this anti-viral factor in primarily protecting epithelial cells and preventing viral invasion through skin or mucosal surfaces rather than in the CNS [27,63,64].

The gradual increase in miRNA-155 expression up to 24 hpi suggests an early inflammatory response, as seen in previous studies [32]. In comparison, ZIKV_MR766_ infection leads to earlier and more significant upregulation of miRNA-155 at 12 and 24 hpi. This is likely due to the higher infectious capacity of ZIKV_MR766_, which may contribute to the stronger host inflammatory response induced by this strain in both in vitro and in vivo experimental models. Notably, miRNA-155 can downregulate host factors SHIP1 and SOCS1, which are responsible for anti-inflammatory and anti-innate immune responses, respectively [65,66,67,68,69]. BV2 cells infected with Japanese encephalitis virus (JEV), another neurovirulent flavivirus, showed an early peak of *ship1* gene expression, followed by an increase in miRNA-155 and subsequent downregulation of SHIP1 [67]. Previous research has suggested that the miRNA-155/SHIP1 axis plays a crucial in microglia stimulation and neuropathology caused by JEV infection, as the downregulation of SHIP1 mediated by miRNA-155 leadss to microglia activation and neuroinflammation [68]. This present study provides new insights into the modulation of SOCS proteins in microglial cells by African and Asian ZIKV strains. Both strains were found to induce expression of the SOCS-1/3 genes in BV2 cells but with different kinetics. ZIKV_PE243_ induced higher expression of SOCS1 in the initial stages of the infection, while there were no significant differences observed in the expression of SOCS3.

Studies have shown that SOCS-1/3 can promote viral survival by negatively regulating signaling pathways of Type I and III IFNs (IFN-λ and IFN-α/β, respectively) [69,70,71]. In addition, SOCS1 has also been shown to inhibit the IFN-γ signaling pathway [72]. The expression profile of SOCS1 and SHIP1 induced by ZIKV_MR766_ and ZIKV_PE243_ is consistent with the downmodulation of miRNA-155 by IL-10 [31]. In dengue virus infection, immune evasion was associated with increased expression of IL-10-mediated SOCS3, which subsequently inactivated JAK/STAT pathways [73]. Furthermore, an imbalance of IL-10 and IL-6, and SOCS1/3 has been observed in patients with severe dengue virus infection [74].

The expression pattern of anti-inflammatory miRNAs 146a and 124, and the inflammatory mediator TRAF6 and IRAK1, suggest an acute increase in inflammation in the early stages of infection, followed by negative feedback. ZIKVV_MR766_ induced a slightly higher expression of miRNA-146a/124 early after infection, indicating the need to counteract the greater inflammation induced by this strain in the very early stages of infection. ZIKV_MR766_ also induced higher expression of TRAF6 early after infection compared to ZIKV_PE243_. The significant drop in the expression levels of TRAF6 at later times of infection, specifically at 24, 48, and 72 hpi, strongly suggests the activation of inhibitory anti-inflammatory mechanisms driven by both anti-inflammatory miRNAs. TRAF6 is associated with positive regulation of inflammatory cytokines, including TNF-α and IL-6, whereas SOCS1/3 may inhibit those mediators. Infection with both ZIKV strains resulted in early stimulation of both pro-inflammatory TRAF6 and anti-viral *socs1* and *ship1* genes. However, a detailed comparison between the ZIKV strains revealed that ZIKV_MR766_ induced a higher expression of the inflammatory gene *traf6* and a lower expression of anti-inflammatory and anti-viral *socs1/ship1* genes.

After infecting microglial cells with Brazilian and African ZIKV strains, the expression of the nuclear receptor PPAR-γ was found to be differentially regulated and may function as an anti-inflammatory mediator. BV2 microglial cells infected with ZIKV_PE243_ showed increased levels of PPAR-γ and decreased levels of IL-6 and TNF-α. Treatment with the PPAR-γ agonist Rosiglitazone significantly reduced the expression of these cytokines, supporting the involvement of this receptor in their regulation. These findings are in line with previous research showing that thiazolidinediones (TZDs) have a neuroprotective effect by inhibiting microglial activation and the expression of chemokines and inflammatory cytokines, including TNF-α and IL-6 [75]. Upon increasing PPAR-γ expression levels, an anti-inflammatory state may be induced, promoting microglial polarization to the tolerant M2-immunosuppression phenotype [39,43] and reducing the production of inflammatory cytokines such as IL-6, TNF-α, and IL-1β, as well as factors such as NF-κβ, reactive oxygen species (ROS) and NO [75]. Our findings are consistent with Sin Foo and colleagues’ observations on the role of monocyte-M1-mediated inflammation in African lineage ZIKV infection and monocyte-M2-mediated immunosuppression in Asian lineage ZIKV infection. Additionally, our results support the previous observation of a higher viral load, and lower IL-10 expression in the African ZIKV strain [76]. We also noted an inverse correlation between miRNA-146a and PPAR-γ expression levels, similar to previous studies on the bacterium *Clostridium difficile*, where elevated miRNA-146a levels reduce the expression of nuclear receptor coactivator-4 (NCOA4), a known activator of PPAR-γ, and of PPAR-γ itself [77]. However, further experiments are needed to establish a cause–effect relationship. Notably, we also found increased levels of PPAR-γ transcripts in ZIKV-infected hNPCs [78]. A recent study found that ZIKV infection in hNPCs resulted in reduced levels of PPAR-γ mRNA ls and NCOA1, coactivators of both RXR and PPAR-γ nucellar receptors [79]. Interestingly, the study also observed an increase in RXR, a positive regulator of PPAR-γ, as well as two negative regulators, FGR and the AP1 transcription factor c-Jun, which appears contradictory. This highlights the complex nature of signaling alterations caused by ZIKV infection. PPARs are known to be involved in cellular pathways related to lipid metabolism, cholesterol content, and biogenesis of peroxisomes. Notably, peroxisomes are crucial sites for some viruses’ biosynthesis, and nonstructural proteins from flaviviruses, including WNV, DENV, and ZIKV, are recruited to their membranes [80,81]. The importance of peroxisomes for ZIKV infection is further supported by impaired virus replication in peroxisome-deficient cells [81]. Individuals infected with ZIKV have been found to have increased levels of several phosphatidylethanolamine lipid species, mainly plasmalogens, which are abundant in the neural membranes of the brain [82]. Plasmalogen biosynthesis requires functional peroxisomes, so it is plausible to hypothesize that the upregulation of PPAR-γ by ZIKV could lead to increased circulation of these phospholipids. Another pathway in which PPAR-γ may play a relevant role in the pathophysiology of the ZIKV infection is the PPAR-γ/mTOR signal pathway [83]. Downmodulation of the Akt-mTOR pathway either by NS4A or NS4B ZIKV protein or by increased PPAR-γ levels may inhibit neurogenesis, induce autophagy, and suppress the expression of the inflammatory cytokines TNF-α and IL-1β [83,84]. Therefore, the ZIKV_PE243_ strain-mediated higher PPAR-γ levels expression may act synergistically with NS4A/NS4B viral proteins inhibiting the Akt-mTOR pathway and contributing to the physiopathology characteristic of this lineage. The observed differences in PPAR-γ expression levels between the two ZIKV strains may explain, at least in part, the differential induction of anti-inflammatory mechanisms and could impact the viral fitness by forming peroxisomes, remodeling host-cell lipid populations, and controlling autophagy and cell survival.

These data strongly suggest that the ZIKV_MR766_ exhibits higher infectivity and induces more inflammation in the early stages of infection. The inflammatory profile induced by each ZIKV strain may be attributed to a delicate balance between pro- and anti-inflammatory miRNAs, their targets, and other anti-inflammatory factors such as PPAR-γ, which are regulated in a time-dependent manner. In addition, the differences observed between the two strains could also be due to intrinsic differences in their viral genomes. These differences could affect receptor usage, specific properties and functions of viral proteins, and the expression of subgenomic flaviviral RNAs (sfRNA) [85,86]. sfRNA can dysregulate RNA decay pathways, suppress the RNAi machinery, and bind to cellular RNAs important for anti-viral responses, thereby impacting the viral cytopathic and pathologic effects [87,88].

One of the limitations of our study is the use of a murine microglial cell line. However, this cell line has been previously used in other flavivirus infection models [67], suggesting that our findings contributed to the understanding of microglial inflammatory and immune responses to viral infections.

Overall, our study highlights significant differences in the inflammatory and anti-viral response of microglial cells to African and Asian lineages of ZIKV, particularly in the induction of miRNAs, inflammatory cytokines, and PPAR-γ expression. The Brazilian ZIKV_PE243_ strain appears to exhibit characteristics of a less virulent stealth virus, as it induces a less robust inflammatory response compared to the African ZIKV_MR766_ strain. Nonetheless, our findings underscore the importance of monitoring circulating African ZIKV to detect new outbreaks, given the neuroinvasive and neurovirulent properties of the ZIKV_MR766_ strain. Moreover, our results offer new insights into the underlining mechanisms governing innate immune regulation of ZIKV_PE243_ infection, which are essential for understanding the pathogenesis of ZIKV-associated diseases and developing therapeutic strategies that can leverage innate immunity to control ZIKV infection.

## Figures and Tables

**Figure 1 viruses-15-01250-f001:**
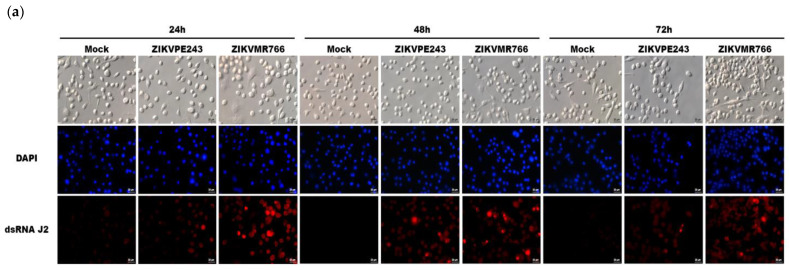
The microglial BV2 cell line is poorly susceptible to infection by African and Asian ZIKV strains. (**a**) BV2 cells were mock-treated or infected with ZIKV_PE243_ or ZIKV_MR766_ (MOI = 10), produced in C6/36 cells. After 24, 48, and 72 hpi, cells were stained with anti-dsRNA antibody, followed by anti-mouse IgG conjugated to PE and analyzed by immunofluorescence. (**b**,**c**) The ZIKV_PE243_ and ZIKV_MR766_ replication kinetics in BV2 cells were determined at different time points and with different MOIs in the pellet by RT-qPCR, as indicated in the figures. (**d**) Titration of infectious particles from BV2 infected cells supernatants was performed in Vero cells by plaque assay. Data are represented as mean ± SD of two independent experiments. (**e**,**f**) Cell viability XTT metabolization assays were carried out in BV2s mock-treated, 1% triton-treated or infected by ZIKV_PE243_ or ZIKV_MR766_ (MOI of 0.1, 1, and 5) produced in C6/36 cells. Data are represented as mean ± SD of three independent experiments and normalized according to the values obtained in cell cultures maintained in culture medium only.

**Figure 2 viruses-15-01250-f002:**
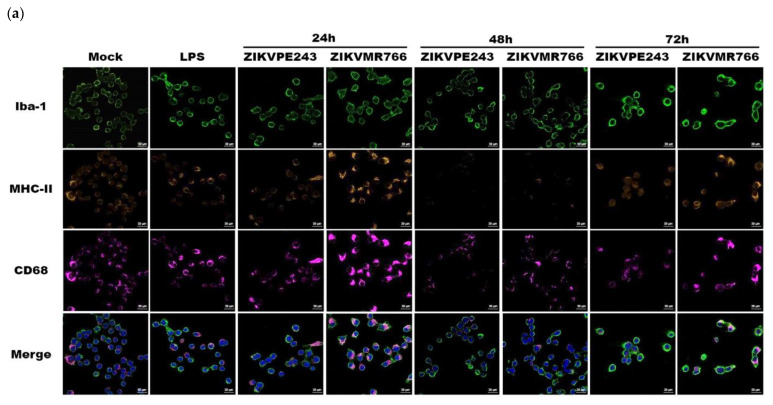
African ZIKV strain induces greater BV2 microglial cell activation. (**a**) BV2 microglial cells were mock-treated, LPS + ATP-treated, or infected with ZIKV_PE243_ or ZIKV_MR766_ with an MOI of 1, produced in C6/36 cells, and evaluated at 24, 48, and 72 hpi. Iba1, MHC-II, and CD68 expressions were determined by immunofluorescence using AlexaFluor488 anti-Iba1, PE-conjugated anti-MHCII, and AlexaFluor647 anti-CD68 and DAPI core marker (blue). (**b**) CD68 mRNA expression levels were measured by qPCR in ZIKV_PE243_ and ZIKV_MR766_ BV2 infected cells with an MOI of 1 at the indicated time points. Bars indicate ∆∆Ct values, normalized according to hprt1 values and mock results. Data are represented as mean ± SD of three independent experiments. Statistical analysis was performed by One-Way ANOVA, considering * *p* < 0.05, ** *p* < 0.01, and *** *p* < 0.001 as significant differences.

**Figure 3 viruses-15-01250-f003:**
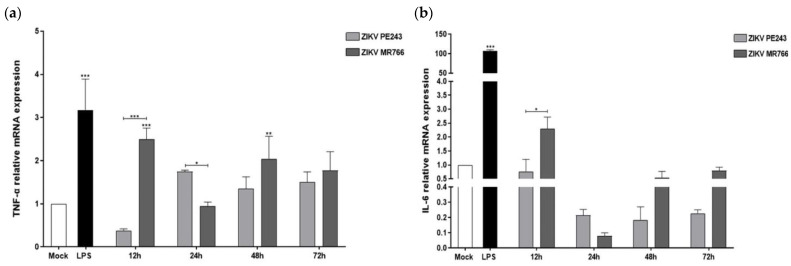
African and Asian ZIKV strains induce differential expression of pro- and anti-inflammatory cytokines and anti-viral factors in BV2 microglial cells. mRNA expression levels were measured by RT-qPCR in ZIKV_PE243_ and ZIKV_MR766_ BV2 infected cells with an MOI of 1 at the indicated time points. (**a**) TNF-α, (**b**) IL-6, (**c**) IFN-γ, (**d**) IL-10, (**e**) IFN-β. Bars indicate ∆∆Ct values, normalized according to hprt1 values and mock values for mRNAs. Data are represented as mean ± SD of three independent experiments. Statistical analysis was performed by One-Way ANOVA, considering * *p* < 0.05, ** *p* < 0.01, and *** *p* < 0.001 as significant differences.

**Figure 4 viruses-15-01250-f004:**
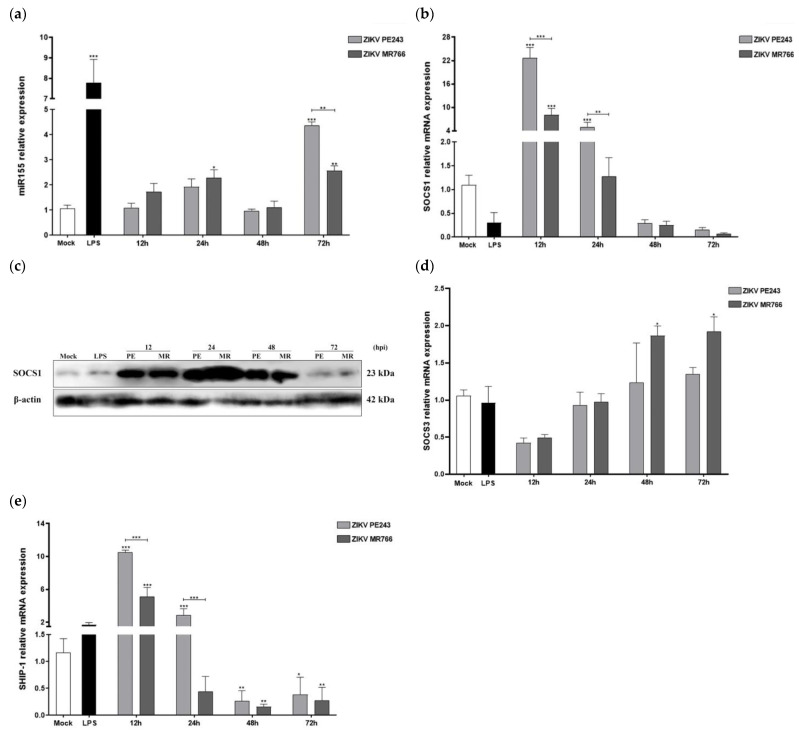
Infection of BV2 microglia cells with ZIKV_PE243_ and ZIKV_MR766_ strains differentially modulate the pro-inflammatory miRNA-155 and anti-inflammatory and anti-viral factors. BV2 cells were infected with both ZIKV_PE243_ or ZIKV_MR766_, with an MOI of 1, at 12, 24, 48, and 72 hpi using mock-treated (Mock) as a negative control and LPS + ATP (LPS) as a positive control. (**a**) miRNA-155, (**b**) SOCS1, (**d**) SOCS3 and (**e**) SHIP1 mRNA expression levels were measured by RT-qPCR. ∆∆Ct plotted results were normalized to hprt1 and mock values for mRNAs, and U6 and mock for miRNAs. (**c**) The protein SOCS1 expressions were analyzed by Western analysis. The membrane was stained with anti-SOCS1 and anti-β-actin as a loading control. Statistical analysis was performed by One-Way ANOVA, considering * *p* < 0.05, ** *p* < 0.01, and *** *p* < 0.001 as significant differences.

**Figure 5 viruses-15-01250-f005:**
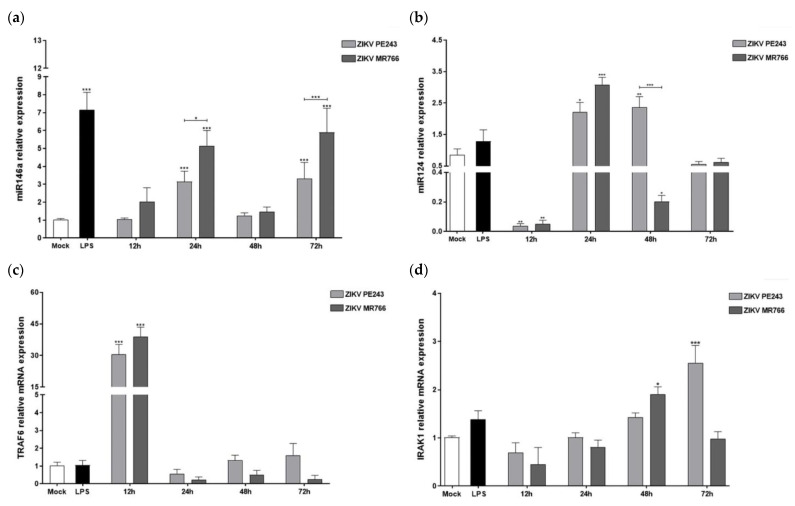
Infection of BV2 microglia cells with ZIKV_PE243_ or ZIKV_MR766_ strains differentially modulates anti-inflammatory miRNAs and inflammatory factors. BV2 cells were infected with both ZIKV_PE243_ or ZIKV_MR766_, with an MOI of 1, at 12, 24, 48, and 72 hpi using mock-treated (Mock) as a negative control and LPS + ATP (LPS) as a positive control. (**a**) miRNA-146a, (**b**) miRNA-124, (**c**) TRAF6 and (**d**) IRAK1. The mRNA expression levels were measured by RT-qPCR. ∆∆Ct plotted results were normalized to hprt1 and mock values for mRNAs, and U6 and mock for miRNAs. Statistical analysis was performed by One-Way ANOVA, considering * *p* < 0.05, ** *p* < 0.01, *** *p* < 0.001, as significant differences.

**Figure 6 viruses-15-01250-f006:**
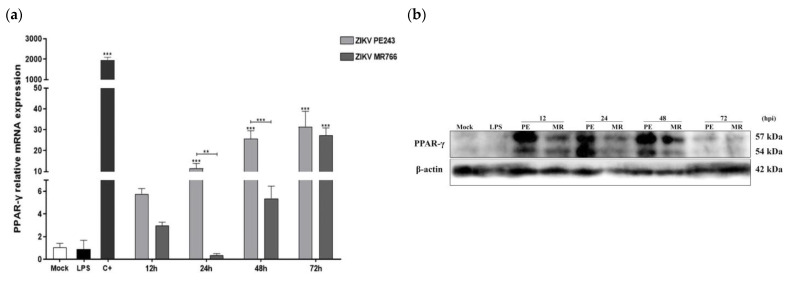
Infection of BV2 microglia cells with ZIKV_PE243_ or ZIKV_MR766_ strains differentially modulates PPAR-γ and their targets. (**a**) PPAR-γ mRNA copy levels were measured by RT-qPCR in BV2 infected cells by ZIKV_PE243_ or ZIKV_MR766_ with an MOI of 1 at 12, 24, 48, and 72 hpi using mock-treated (Mock) and LPS + ATP (LPS) as negative controls, and adipocyte mice cells (C+) as positive controls. (**b**) The PPAR-γ protein expression levels were analyzed by Western blot analysis. The membrane was stained with anti-PPAR-γ and anti-β-actin as a loading control. (**c**,**d**) IL-6 and TNF-α mRNA copy levels were measured by RT-qPCR. BV2-infected cells were evaluated at 48 hpi and treated with PPAR-γ agonist-Rosiglitazone. As negative controls were used, mock-treated with agonist (M+Rosi) and mock-treated without agonist (-Rosi), the positive control was mock-treated with LPS. Data are represented as mean ± SD of three independent experiments. Statistical analysis was performed by One-Way ANOVA, considering * *p* < 0.05, ** *p* < 0.01, and *** *p* < 0.001 as significant differences.

**Table 1 viruses-15-01250-t001:** Comparison of inflammatory, anti-inflammatory, and anti-viral marker expressions by BV2 microglial cell line after infection with Brazilian and African ZIKV strains (ZIKV_PE243_ and ZIKV_MR766_).

	12 hpi		24 hpi		48 hpi		72 hpi	
	PE243	MR766	PE243	MR766	PE243	MR766	PE243	MR766
**Inflammatory Markers**								
TNF-α	0.37	**2.49**	1.75	0.95	1.35	**2.04**	1.50	1.77
IL-6	0.77	2.30	0.22	0.08	0.18	0.55	0.23	0.80
IFN-γ	-	-	1.68	1.26	2.03	**6.83**	**7.36**	**11.05**
miRNA155	1.02	1.63	1.81	2.14	0.92	1.05	**4.10**	2.41
TRAF6	**30.02**	**38.23**	0.55	0.22	1.30	0.48	1.56	0.25
IRAK1	0.69	0.45	1.00	0.80	1.42	**1.89**	**2.53**	0.97
**Anti-Inflammatory Markers**								
IL-10	1.04	0.89	**3.12**	1.75	1.55	1.38	1.50	**3.52**
PPAR-γ	5.53	2.84	**11.05**	0.37	**24.65**	5.15	**30.13**	**26.33**
miRNA146a	1.01	1.98	**3.04**	**4.99**	1.21	1.41	**3.23**	**5.72**
miRNA124	**0.04**	**0.06**	**2.63**	**3.65**	**2.80**	**0.24**	0.66	0.73
SHIP1	**8.98**	**4.39**	**2.44**	0.38	**0.23**	**0.14**	**0.33**	**0.24**
**Anti-Viral Response**								
SOCS1	**20.68**	**7.34**	**4.51**	1.17	0.27	0.23	0.14	0.06
SOCS3	0.40	0.46	0.88	0.92	1.17	**1.76**	1.28	**1.82**
IFN-β	0.75	0.35	**3.07**	1.22	**1.86**	1.62	1.34	1.16

## Data Availability

The data presented in this study are available in the article and are available on request from the corresponding author.

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
