# Peer review of "ZIKV Strains Elicit Different Inflammatory and Anti-Viral Responses in Microglia Cells"

_viruses, 2023, doi:10.3390/v15061250_

Round 1

Reviewer 1 Report

In this study, it was compared differential expression miRNAs 25 (miRNA-155/146a/124) and their cellular targets (SOCS1/3, SHP1, TRAF6, IRAK1), as well as pro- 26 and anti-inflammatory and antiviral cytokines (IL-6, TNF-a, IFN- γ, IL-10, and IFN-β) in BV2 a microglia cells infected with two ZIKV strains from African and Asian lineages. As a result the article is very interesting and revealing. The authors verified the ZIKV-mediated modulation of inflammatory and anti-viral innate immune responses in cells derived of murine origin and infering how viral infection would be in human cells.

I have questions for the authors. 

According to the text and cited by the authors : Lines 269/270 the MOI 1 is cited as “barely detected when BV2 were infected at MOI 1 (data not shown)” and in Lines 285/286 refers this MOI as is “more physiological context” and all the experiments were done by the authors in these viral infection conditions. My question is about MOI 1:  Is this MOI good enough to infected all cells or at least to guarantee the viral infection in all cells ? and to inititate all experiments?.  (Apparently not). 

Author Response

Response:

We are aware that it is virtually not possible to achieve 100% viral infection in a cell culture, using an MOI of 1, and the MOI to be chosen will depend on the biological issue that is being investigated. In fact, we accessed BV2 infection with ZIKV isolates using different MOIs and distinct techniques, including RT-qPCR, dsRNA staining and IF, and plaque assay, but we hadn’t included all the assays in the first version of the MS for simplification. However, given the reviewers comments, we agree that those experiments should be included in the final version to clarify this in vitro model.

At first, we could clearly detect the presence of dsRNA upon infection of both virus strains at a high MOI (10), indicating intracellular virus replication, although it could not be clearly evidenced with an MOI of 1. Then, we evaluated the infection with using the MOIs of 0.1, 1 and 5 by RT-qPCR, which is more sensitive than IF. In those experiments, analysis of intracellular viral RNA showed that infection with MOIs of 0.1 and 1 resulted in typical replication curve with an eclipse phase detected at 8hpi, followed by an increased in RNA concentration until 48 hpi. On the other hand, when the viruses were inoculated at a MOI of 5, we could not detect a clear eclipse nor a significant increase of intracellular vRNA concentration at later time points. In addition, similar RNA concentration was detected after infection with MOIs of 1 and 5 at 24 hpi., what might indicate a replication threshold in this system (Figure 1c, 1c and 1d of the revised MS). To investigate whether infection of BV2 resulted in a productive viral replication, the concentration of released infectious particles were measured in the supernatants by plaque assay. Virus inoculation at a MOI of 0.1 resulted in very low to undetectable virus titers (<102; data not shown). But when higher MOIs were used, significant increase in the PFU released in the supernatants was detected at 24hpi, evidencing a productive replication of both virus strains in BV2 cells. Infection with ZIKVMR766 resulted in higher PFU concentration, in comparison to ZIKVPE243, at both MOIs. Importantly, when the cells were infected with an MOI of 5, but not at an MOI 1, a decreased in PFU was detected after 48hpi (corroborating the RT-qPCR data).

Therefore, although an MOI of 1 will certainly not guarantee that all cells will be infected, this concentration may better represent a physiological kinetics of infection, where the infection starts with a discrete amount of virus and progresses over time. It also allowed the detection of subtle differences in the induction of inflammatory and antiviral markers, between both strains; what could be jeopardized by a higher MOI. Therefore, we chose a MOI of 1 to proceed with the analysis of the biological effect of ZIKV in BV2 cells.

Figure 1 and the corresponding text were completely reformulated in order to include these data.

Reviewer 2 Report

In this manuscript, author did the comparative expression analysis of miRNAs (miRNA-155/146a/124) and their cellular targets (SOCS1/3, SHP1, TRAF6, IRAK1), as well as pro and anti-inflammatory and antiviral cytokines (IL-6, TNF-, IFN- γ, IL-10, and IFN-β) in BV2 microglia cells infected with ZIKV strains derived from African and Asian lineages. Author also showed that the ZIKKPE243 strain induced significantly higher levels of the anti-inflammatory nuclear receptor – PPAR-γ and the infection with the ZIKVMR766 strain promoted both a higher inflammatory response and a lower antiviral factors expression.  It is clearly a big effort of the author to do this study, but despite all the positive outcomes, I have some suggestions where it can be improved.

Minor Comments

1.     Line spacing is not the same all over the paper, the author used Palatino Linotype font at some area of paper and at some area Calibri font has been used. Please be consistent throughout the paper.

2.     In line 126, Author used (64) for reference, but throughout the paper author used [ ] for reference. Please be consistent throughout the paper.

3.     In line 367-370, the Author gave space between 24 and hpi, but somewhere there is no space. Please be consistent throughout the paper.

Major Comments

1.     In figure 1a author infect cells with 10 MOI, saying that these strains were not infecting at lower MOI but after that in figure 1 and 2 cells were infected with 1MOI. Please explain.

2.     In figure 1c, ZIKV PE243 infection at 24hpi is higher as compared to 0hpi whereas after 72hpi infection is decreased, please explain.

3.     In figure 3a, ZIKVMR766 infection stimulated higher TNF-α levels than mock-infected cells at 12hpi and the level is lower at 48hpi and again the level of TNF-α is higher at 72hpi, please explain.

4.     In figure 4a, ZIKV infection led to a gradual increase in miRNA-155 expression between 12 and 24hpi, followed by a decline at 48hpi, and a subsequent significant increase at 72hpi. Please explain.

5.     Another limitation of this study, microglial cell line used in this study was of murine origin. Can we co-relate these inflammatory anti-viral responses to humans, please explain.

It is clearly a big effort of the author to do this study, but despite all the positive outcomes, Minor editing of English language is required.

Author Response

Minor Comments

  1. Line spacing is not the same all over the paper, the author used Palatino Linotype font at some area of paper and at some area Calibri font has been used. Please be consistent throughout the paper.

Response: We checked all the text and standardized the font to Palatino Linotype. For the spacing we followed the model provided by Viruses, in this model there are different spacing at some points (as in the division between topics). Other than that, we had to make some adjustments to the spacing so that the images and the letters that indicate them don't stay separated and the table doesn't stay divided into two pages and impairs the reading.

In addition, we noticed that there was some change in the doc manuscript that we submitted (possibly an update that went unnoticed), since the spacing and layout of the doc that we receive and the PDF that we submitted were different. We fixed it and hope this doesn't happen again.

  1. In line 126, Author used (64) for reference, but throughout the paper author used [ ] for reference. Please be consistent throughout the paper.

Response: We fixed the error.

  1. In line 367-370, the Author gave space between 24 and hpi, but somewhere there is no space. Please be consistent throughout the paper.

Response: We check the entire text and standardize the space.

Major Comments

  1. In figure 1a author infect cells with 10 MOI, saying that these strains were not infecting at lower MOI but after that in figure 1 and 2 cells were infected with 1MOI. Please explain.

Response: In fact, we had initially performed additional molecular assays with different MOIs which were not included in the first version of the MS, for simplification. However, given the comments of the reviewers, we decided to include them in order to better demonstrate BV2 infection model. Although dsRNA staining was not clear detected when the cells were infected with MOI of 1, RT-qPCR and plaque assay, which are more sensitive, had shown a typical replication curve and release of infectious particles when both viruses were inoculated at this MOI. In addition, similar RNA concentration was detected after infection with MOIs of 1 and 5 at 24 hpi., what might indicate a replication threshold in this system. Therefore, we chose a MOI of 1 to proceed with the analysis of the biological effect of ZIKV in BV2 microglial cells, in a more natural context of infection, in order to guarantee the detection of subtle differences in the induction of inflammatory and antiviral markers, between both strains.

We completely reformulated Figure 1 system (Figure 1c, 1c and 1d of the revised MS) and the corresponding text to include those data.

  1. In figure 1c, ZIKV PE243 infection at 24hpi is higher as compared to 0hpi whereas after 72hpi infection is decreased, please explain.

Response: Indeed, infection of BV2 with ZIKVMR766 was more efficient, resulting in continuous increase of released virus particles, whereas infection with ZIKVPE243 resulted in an initial release of PFU, with a later decrease, what might indicate a higher concentration of immature particles in the supernatants due to lower virus replication efficiency. However, this was not directly addressed in this MS.

  1. In figure 3a, ZIKVMR766 infection stimulated higher TNF-α levels than mock-infected cells at 12hpi and the level is lower at 48hpi and again the level of TNF-α is higher at 72hpi, please explain.

Response: Virus replication cycles includes sequential events, that take hours to be completed (8-10 hours), and interfere with several cellular metabolism pathways, like the ones involved in genome sensing, protein synthesis and degradation, and energy metabolism, for example. All this may be sensed and counteracted by other regulatory events, such as the ones investigated here – miRNA regulation and cytokine production. Therefore, it is expected that those events happen in temporal waves, which extent will depend on the virus biochemistry and host cell type. In this sense, and as indicated by the reviewer, ZIKVMR766 induced increased expression of the proinflammatory cytokines TNF-a and IL6, followed by a decrease at 24hpi and a further increase at later time points. Interestingly, miR146a and miR124 were upregulated at 24hpi (which was also associated with TRAF-6 inhibition), and then decreased at 48hpi. One can then speculate that miRNA and cytokine expression pathways are modulating each other during the infection cycles course. Although not directly accessed here, we believe that further analysis of each regulatory step based on molecular gene depletion or expression worth further investigation.

  1. In figure 4a, ZIKV infection led to a gradual increase in miRNA-155 expression between 12 and 24hpi, followed by a decline at 48hpi, and a subsequent significant increase at 72hpi. Please explain.

Response: As discussed in the previous response, we believe that a complex interplay between cellular metabolic pathways and virus replication steps may happen in waves during replication cycles. As pointed by the reviewer, miR155 (which is usually activated in the initial stage of the innate response) is enhanced at earlier time points, and then downregulated at 48hpi. A similar pattern was also detected regarding miR146a and miR124. On the other hand, expression of IFN-g and IL-10 are detected at later time points and may be associated to diminished expression of certain miRNAs, such as miRNA-155, which was addressed in the MS discussion.

  1. Another limitation of this study, microglial cell line used in this study was of murine origin. Can we co-relate these inflammatory anti-viral responses to humans, please explain.

Response: Microglia activation had been previously detected in a mouse model of ZIKV infection, what could be related to enhanced TNF-a levels in the brain and was associated with engulfment of hippocampal presynaptic terminals and memory impairment (doi:10.1038/s41467-019-11866-7). In addition, our findings are supported by other studies, in which it is shown that human microglia cells can contribute to viral reservoir and source of inflammation in the CNS (https://doi.org/10.1093/hmg/ddx382) (https://doi.org /10.1093/cid /ciw878). Moreover, the greater African strain virulence, along with its greater ability to induce inflammation than the Asian strain is in line with other studies, carry out with different human cell types of CNS (http://dx.doi.org/10.1016/j.ebiom .2016.09.020) (doi:10.3389/fmicb.2017.02557) (https://doi.org/10.1128/JVI.00640-19).

An important finding of our work to highlight, and not previously described in human, is that the ZIKVPE243 strain induced significantly higher levels of the anti-inflammatory nuclear receptor – PPAR-γ than the ZIKVMR766. The higher ZIKVPE243 strain-mediated PPAR-γ expression may play an important role in the lower inflammation induced by the Asian strains and contributing to a passive strategy of innate immune evasion. Therefore, our study paves the way for future investigations, in order to evaluate the PPAR-γ expression by other Asian strains, and in different human cells of CNS.

In summary, these finding emphasize the possibility that different ZIKV strains might also display the same phenotypic differences in humans.

Comments on the Quality of English Language

It is clearly a big effort of the author to do this study, but despite all the positive outcomes, Minor editing of English language is required.

Response: The requested English corrections were made.

Round 2

Reviewer 2 Report

The author made a great effort, overall, the manuscript has been modified extensively.  I am  accepting this paper in present form.